# FROM HUMAN HANDS TO ROBOT ARMS: MANIPULATION SKILLS TRANSFER VIA TRAJECTORY ALIGNMENT

## ABSTRACT

Learning diverse manipulation skills for real-world robots is severely bottlenecked by the reliance on costly and hard-to-scale teleoperated demonstrations. While human videos offer a scalable alternative, effectively transferring manipulation knowledge is fundamentally hindered by the significant morphological gap between human and robotic embodiments. To address this challenge and facilitate skill transfer from human to robot, we introduce **Traj2Action**, a novel framework that bridges this embodiment gap by using the 3D trajectory of the operational endpoint as a unified intermediate representation, and then transfers the manipulation knowledge embedded in this trajectory to the robot's actions. Our policy first learns to generate a coarse trajectory, which forms an high-level motion plan by leveraging both human and robot data. This plan then conditions the synthesis of precise, robot-specific actions (e.g., orientation and gripper state) within a co-denoising framework. Extensive real-world experiments on a Franka robot demonstrate that Traj2Action boosts the performance by up to **27%** and **22.25%** over $\pi_0$ baseline on short- and long-horizon real-world tasks, and achieves significant gains as human data scales in robot policy learning. Our project website, featuring code and video demonstrations, is available at https://anonymous.4open.science/w/Traj2Action-4A45/.

## 1 INTRODUCTION

Enabling robots to master a diverse array of manipulation skills in the real world presents a formidable challenge, primarily due to the data-hungry nature of modern policy learning (Kim et al., 2024; Black et al.; Team et al., 2024). The prevailing paradigm of imitation learning relies on large-scale datasets of expert demonstrations (O'Neill et al., 2024; Khazatsky et al., 2024), which are typically gathered through time-consuming teleoperation (Wu et al., 2023; Fu et al., 2024; Zhao et al., 2023; Orbik, 2021). This process not only requires significant investment in specialized hardware but also demands extensive operator training, rendering it a critical bottleneck for scaling up robotic capabilities.

While human videos offer a cost-effective and abundant alternative data source, their direct application is fundamentally hindered by the significant morphological gap between embodiments, as they lack the robot-specific action labels required for direct mimicry. Prior works have attempted to learn a shared visual representation across both human and robot videos (Wang et al., 2023), formulating reward functions (Ma et al., 2022; Shao et al., 2021; Ma et al., 2023), or using human demonstrations as high-level context for robot action prediction (Zhu et al., 2025; Shah et al., 2025). However, these indirect approaches often fail to leverage the rich, explicit motion signals embedded within human actions. More direct lines of research attempt to define a unified action space or to transfer motion skills in a way that can be effectively utilized by robots. Yet, these strategies often introduce their own limitations. Some methods adopt a two-stage pipeline Bharadhwaj et al. (2024); Bi et al. (2025); Luo et al. (2025), where models are first trained on human data and then adapted with robot data, but this sequential procedure can constrain the final quality of robot policy. Other joint-training approaches (Yang et al., 2025; Qiu et al., 2025) are typically confined to kinematically similar embodiments like humanoid robots, while methods (Ren et al., 2025; Kareer et al., 2025) that directly align mismatched poses (e.g., human hand to parallel gripper) struggle to establish a physically meaningful correspondence. Even though some works (Bharadhwaj et al., 2024) adopt simplified representations like rigid transformations to mitigate the embodiment gap, fundamental structural differences mean that the physical interpretation of rotations remains misaligned. Furthermore, other

methods (Park et al., 2025; Liu et al., 2025) impose heavy constraints, either requiring strictly paired human-robot demonstrations or relying on complex motion retargeting pipelines, which restricts their applicability and negates the cost-benefit of using human data.

To address these limitations, we propose Traj2Action, a framework that transfers motion knowledge from human to robot learning process by leveraging 3D trajectories as a robust intermediate representation. Our method follows a coarse-to-fine action prediction paradigm for robot, where coarse trajectory planning derived from both human and robot demonstrations guides fine-grained robot action learning. Specifically, we introduce a unified trajectory representation of human hand and robot end-effector 3D positions, which reduces embodiment discrepancies and allows abundant, cost-effective human data to improve coarse trajectory planning. We then utilize a co-denoising training scheme over trajectories and actions, which enables the policy to focus on generating robot actions with precise orientation and gripper states under the guidance of the coarse trajectory plan. To further bridge the observational gap, we designed a hand wrist-mounted camera to supply ego-view for human data collection, which mimics the robot's ego-centric view observation and thus enhances cross-embodiment training effectiveness and consistency.

By leveraging human demonstrations, our method surpass a traditional VLA baseline on short- and long-horizon real-world tasks by a large margin. Besides, as human data scales, our method shows a significant performance gain in robot manipulation tasks. Furthermore, we show that our method allows for replacing a significant amount of expensive robot data with low-cost human demonstrations while achieving comparable final policy performance, validating the effectiveness of using low-cost human demonstration data for robot learning.

## 2 RELATED WORKS

**Robotic Imitation Learning**. Imitation learning (IL) (Schaal, 1996; Pomerleau, 1988; Atkeson & Schaal, 1997) has emerged as a dominant paradigm for tackling complex robotic manipulation tasks, enabling agents to learn expert behaviors from visual observations and language instructions. Recent years have witnessed remarkable progress in this domain, largely propelled by sophisticated policy architectures and novel training schemes like Action Chunking Transformer (ACT) (Zhao et al., 2023), Diffusion Policy series (Chi et al., 2023; Reuss et al., 2024). These policies excel at mapping high-dimensional sensory inputs and proprioceptive states to robot actions.

Despite their success, the performance of these state-of-the-art methods is fundamentally contingent upon access to large-scale, expert-level robot demonstration datasets (Khazatsky et al., 2024; O'Neill et al., 2024). This dependency has co-evolved with the rise of generalist Vision-Language-Action Models (Black et al.; Kim et al., 2024), which are trained on these massive datasets with the goal of creating a single, generalizable policy that spans diverse tasks and environments. While significant efforts have been made to streamline the burdensome process of teleoperated data collection—through innovations such as kinesthetic teaching (Wu et al., 2023) and VR-based mirroring (Ding et al., 2024; Shaw et al., 2024; Park & Agrawal, 2024)—a critical bottleneck persists. Expanding these datasets to cover a long tail of diverse tasks and environments remains prohibitively challenging and expensive. This fundamental limitation motivates the search for alternative paradigms capable of learning effective policies from more abundant, scalable, and lower-cost data sources.

**Learning from Human Demonstration**. Given the challenges associated with collecting teleoperated robot data, videos of human activity have become a popular alternative data source due to their vast availability and ease of collection. One line of work utilizes large-scale human video datasets for pre-training powerful visual representations (Nair et al., 2022; Karamcheti et al., 2023; Baker et al., 2022). However, they lack the direct training of action labels from human videos and hinders transferability to robot learning. Another line of research (Bi et al., 2025; Luo et al., 2025) attempt to directly pre-train a robot policy on human videos by using hand poses as action labels, followed by lightweight fine-tuning on the target robot. However, this strategy's fail to address the significant morphological gap between a human hand and a robotic end-effector, which hinders effective skill transfer. Several approaches (Wang et al., 2023; Zakka et al., 2022; Xu et al., 2023) focus on task-specific adaptation, where human videos are used as a reference to enable few-shot learning on new tasks with a small number of robot demonstrations. However, these approaches requires strictly paired human-robot datasets, a constraint that significantly increases data collection difficulty and cost.

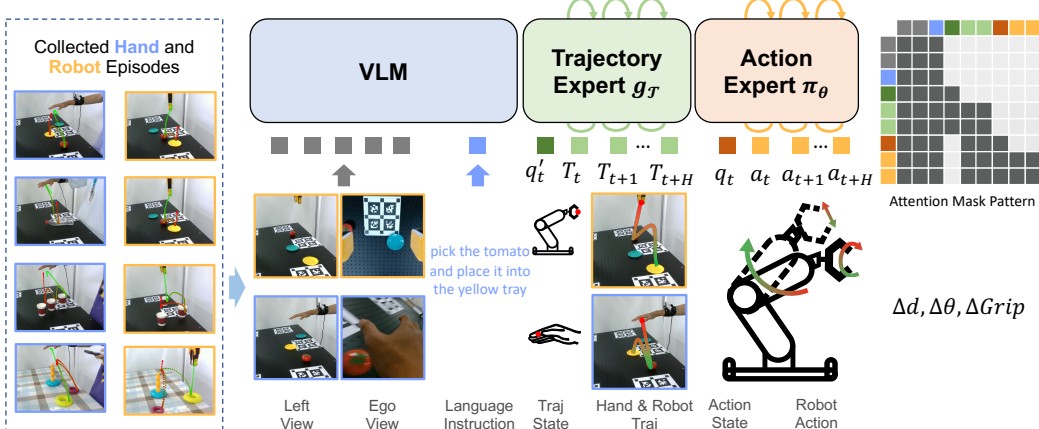

**Figure 1:** An overview of the **Traj2Action** framework. Given multi-view images and a language instruction, the model operates in a coarse-to-fine manner. A Trajectory Expert, trained on both human and robot data, first predicts a coarse 3D trajectory plan. This high-level plan then conditions an **Action Expert** to generate fine-grained robot actions, which include precise translation ($\Delta d$), rotation ($\Delta\theta$), and gripper state ($\Delta\text{Grip}$). Both experts are optimized jointly within a co-denoising framework, enabling the coarse trajectory to guide the synthesis of fine-grained actions.

To specifically address the morphological gap, researchers have proposed several strategies. For kinematically similar embodiments, such as dexterous hands, a common approach is to model both the human and robot hand with a unified 3D mesh and use motion retargeting to translate human keypoints into robot joint positions (Yang et al., 2025; Park et al., 2025; Qiu et al., 2025; Luo et al., 2025). For more heterogeneous pairings, like a human hand and a simple parallel gripper, methods often build a unified representation in an intermediate space. This includes predicting the future pixel locations of 2D keypoint tracks in the image space (Ren et al., 2025) or jointly regressing disparate human and robot end-effector poses (Qiu et al., 2025).

However, these methods often rely on complex, indirect mappings or intermediate representations that may not robustly capture the core task semantics. In contrast, our work seeks a more direct and fundamental correspondence. We posit that the end-effector trajectory—the 3D path of the operational endpoint—is the greatest common divisor that preserves the essential intent of a manipulation task across different embodiments. By abstracting away low-level morphological details, we use this simple yet effective trajectory as a unified action representation, which serves as a powerful manipulation prior for robot policy learning.

## 3 METHOD

In this section, we present **Traj2Action** for transfering manipulation skills from human demonstrations to robot. Our methodology is structured to answer two key questions: (1) *How can we unify human and robot demonstrations despite their significant embodiment differences?* and (2) *How can this unified knowledge be effectively integrated into the robot's learning process?* To this end, we introduce unification strategy for human and robot, which involves representing motions in a unified trajectory space (Section 3.1). We then detail the Traj2Action, which is designed to translate this unified trajectory representation into fine-grained robot actions (Section 3.2). Data collection pipeline for acquiring the necessary cross-embodiment demonstrations is introduced finally (Section 3.3).

### 3.1 UNIFIED TRAJECTORY SPACE

A fundamental challenge in leveraging human demonstrations is bridging the morphological gap between a human hand and a robotic end-effector such as gripper. To address this, we introduce a unified trajectory space that serves as a common ground for policy learning. Formally, we define the trajectory as a sequence of cartesian positions, with each position denoted by $\mathbf{T} \in \mathbb{R}^3$. For a human demonstration, the trajectory is derived from the midpoint of the thumb and index finger keypoints. For a robot demonstration, the trajectory is the 3D position of its end-effector. This

representation abstracts away low-level embodiment details (e.g., finger articulation vs. gripper width) and instead captures the high-level intent of a task by preserving the essential motion of the operational endpoint. By modeling both demonstrations in this shared space, we establish a unified foundation for knowledge transfer.

## 3.2 TRAJ2ACTION

Having established a unified representation, we now introduce the Traj2Action framework, designed to effectively leverage the combined knowledge from both demonstration types and translate it into precise robot actions. As illustrated in Figure 1, the framework consists of a dual-expert policy architecture trained via a joint denoising objective.

**Policy Architecture.** Traj2Action builds upon a pretrained Vision-Language-Action (VLA) backbone, $\pi_0$, which is composed of a Vision-Language Model (VLM) for processing visual and language inputs, and an action expert for generating control signals. We extend this foundation with an additional Trajectory Expert, $g_{\mathcal{T}}$, whose architecture mirrors that of the pretrained action expert and model weights $\mathcal{T}$ initialized by the action expert of $\pi_0$. The Trajectory Expert is responsible for learning the shared, high-level motion prior by training on both human and robot data ($\mathcal{D}_h \cup \mathcal{D}_r$) to predict a coarse future trajectory $\mathbf{T}_{t+1:t+H} \in \mathbb{R}^{H \times 3}$, where $H$ is the prediction horizon. The Action Expert $\pi_\theta$ then translates this high-level spatial plan into fine-grained, robot-specific actions. It learns exclusively from robot data $\mathcal{D}_r$ and is crucially conditioned on the trajectory expert's plan to predict a future action sequence $\mathbf{a}_{t+1:t+H} \in \mathbb{R}^{H \times 7}$. Each action vector $\mathbf{a}_t \in \mathbb{R}^7$ comprises a 3D end-effector position delta, a 3D axis-angle rotation delta, and a 1D gripper state.

**Joint Denoising for Trajectory-Conditioned Action Generation.** We train both experts jointly using a flow matching objective. The Trajectory Expert is trained to denoise a noisy trajectory $\mathbf{T}_\tau = \tau \cdot \mathbf{T}^*_{t+1:t+H} + (1 - \tau)\mathbf{z}$, where $\mathbf{T}^*_{t+1:t+H}$ is the ground-truth trajectory and $\mathbf{z} \sim \mathcal{N}(0, \mathbf{I})$ denotes the gaussain noise, and $\tau \in [0, 1]$ represents the flow time in flow matching. The trajectory denoising loss is formulated as:

$$\mathcal{L}_{\text{traj}}(\mathcal{T}) = \mathbb{E}_{\tau, \mathbf{z}, \mathbf{T}^*} \left[ \| g_{\mathcal{T}}(\mathbf{T}_\tau, \tau, \mathbf{I}_t, \mathcal{P}, \mathbf{q}'_t) - (\mathbf{z} - \mathbf{T}^*_{t+1:t+H}) \|^2 \right], \tag{1}$$

where conditioning variables are the visual observations $\mathbf{I}_t$, language instruction $\mathcal{P}$, and the current trajectory state $\mathbf{q}'_t \in \mathbb{R}^3$ representing 3D position.

The Action Expert is trained to denoise a noisy action $\mathbf{a}_\tau = \tau \cdot \mathbf{a}^*_{t+1:t+H} + (1 - \tau)\mathbf{z}$, conditioned on the noisy action $\mathbf{a}_\tau$ and noisy trajectory $\mathbf{T}_\tau$ with a causal attention pattern as shown in the right of Figure. 1. This conditioning is the key mechanism for integrating the manipulation prior from the corresponding denoising process of unified trajectory. The action denoising loss is calculated by:

$$\mathcal{L}_{\text{action}}(\theta) = \mathbb{E}_{\tau, \mathbf{z}, \mathbf{a}^*} \left[ \| \pi_\theta(\mathbf{a}_\tau, \mathbf{T}_\tau, \tau, \mathbf{I}_t, \mathcal{P}, \mathbf{q}_t) - (\mathbf{z} - \mathbf{a}^*_{t+1:t+H}) \|^2 \right], \tag{2}$$

where $\mathbf{q}_t \in \mathbb{R}^8$ is the full-dimensional robot proprioceptive state, consisting of the 3D end-effector position, its 4D quaternion orientation, and the 1D gripper width. The total loss is $\mathcal{L} = \mathcal{L}_{\text{traj}} + \mathcal{L}_{\text{action}}$. At inference, both outputs are generated in parallel by an ODE solver with same denoising steps, which enables that no extra delay for action prediction.

## 3.3 DATA COLLECTION FOR HUMAN AND ROBOT

To collect the unified trajectories and robot action demonstrations for our cross-embodiment dataset $\mathcal{D} = \mathcal{D}_h \cup \mathcal{D}_r$, we present the data collection systems as follows.

**Human Hand Motion Capture System.** We capture high-fidelity 3D human hand poses using a calibrated multi-camera system (as shown in Figure 2). Our pipeline uses Google MediaPipe (Zhang et al., 2020) to detect 2D hand keypoints from three views, then fits the parametric MANO (Romero et al., 2017) 3D hand model to these keypoints by minimizing reprojection error. This yields an accurate 3D hand motion, from which we extract the operational endpoint's trajectory.

**Robot Data Collection.** Our platform features a Franka Research 3 arm with a UMI Gripper (Chi et al., 2024). An expert teleoperates the robot via a SpaceMouse. We record proprioceptive data and images from a static third-person camera and a wrist-mounted camera. The robot's end-effector 3D position is directly recorded as its trajectory. Simultaneously, the end-effector's 3D orientation and the gripper state are also recorded. Combined with the 3D position, these constitute the full action

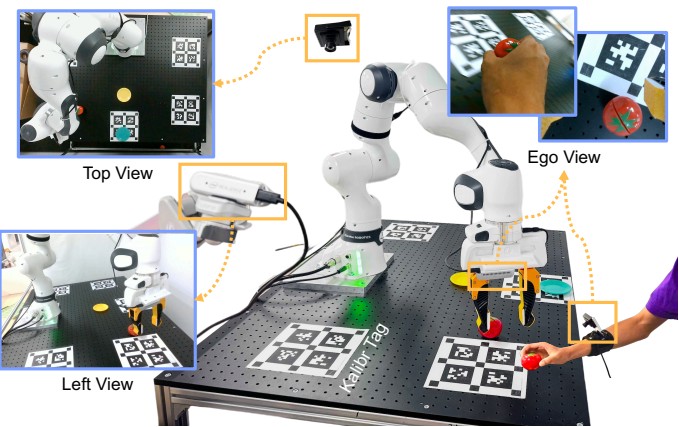

**Figure 2:** Illustration of our data collection systems for human hand motion (left) and robot teleoperation (right).

label for policy training. To enhance policy robustness, we randomize the robot's starting pose for each demonstration, which improves the policy's ability to recover from errors. For more details, please refer to Appendix A.2.

## 4 EXPERIMENTS

**Real-World Tasks.** As shown in Figure 3, four distinct real-world tasks are designed to evaluate the policy's capabilities across basic manipulation, language grounding, long-horizon planning, and precise control. The tasks are detailed as follows:

Task 1: *pick up the water bottle* (PWB): The robot is tasked with locating a water bottle placed on a tabletop, moving towards it, and grasping it successfully. This task evaluates the model's fundamental pick-and-place capabilities. Task 2: *pick up the tomato and put it in the yellow/blue tray* (PTT): The workspace contains a tomato and two trays, one yellow and one blue. The robot must pick up the tomato and place it into the tray specified by a language command (e.g., "the yellow tray"). This task tests the policy's ability to task instructions to specific objects and goals. Task 3: *stack the rings on the pillar* (SRP): The scene includes a pillar (composed of a yellow column and a blue base), a yellow ring, and a red ring. The robot needs to pick up both rings, one by one, and place them onto the pillar. This task assesses multi-step object manipulation and precision location, which demands that the policy has precise action control. Task 4: *stack the paper cups* (SPC): Three paper cups are placed on the table. The robot is required to stack them sequentially to form a single tower. This task evaluates the policy's ability to handle deformable objects and perform iterative, precise placement. We categorize these tasks as either **Short-Horizon** or **Long-Horizon**. Tasks 1 and 2 are Short-Horizon, evaluating fundamental skills like object grasping and language grounding. In contrast, Tasks 3 and 4 are significantly more demanding Long-Horizon tasks that test the policy's planning capability. They not only require the aforementioned skills but also critically assess the policy's ability to perform high-level planning for complex, sequential operations.

**Data Collection Efficiency Analysis.** As presented in Table 1, a summary of our data collection effort reveals a significant disparity in collection efficiency between human hand and robot teleoperation. For each demonstration, we measure both *Collect Time* (the execution duration of the task itself) and *Reset Time* (the period needed to restore the robot, human and objects for the next trial). Overall, Collect Time from a human expert is roughly **3.5** times faster per demonstration than robot teleoperation (an average of 5.09s vs. 17.98s).

This overall efficiency advantage is amplified in more complex, long-horizon tasks, an effect primarily driven by differences in Collect Time. For instance, in the multi-stage *stack the paper cups* task, the human demonstrator's superior dexterity and agility during the collection phase makes them 3.85 times more efficient. In contrast, for the simpler, short-horizon *pick up the tomato and put it in the tray* task, the efficiency gain during collection was a more modest 2.52 times. This trend underscores a key finding of our work: the efficiency benefit of leveraging human data is greatest for the most challenging tasks, offering a scalable path forward for teaching robots complex manipulation skills.

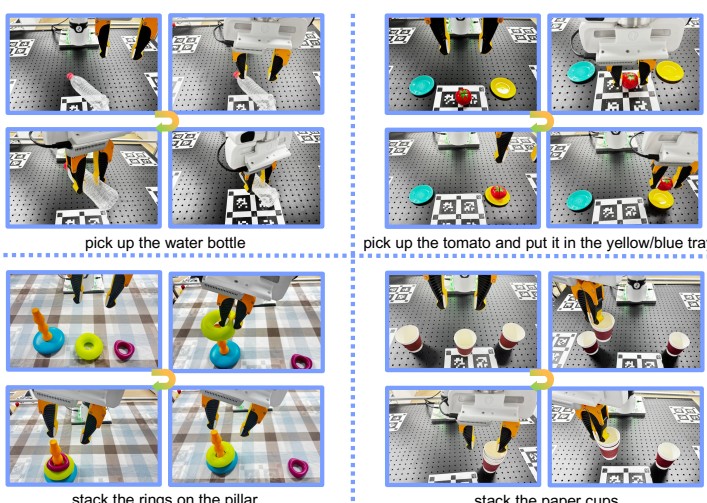

**Figure 3:** Visual illustration of four real-world tasks in Franka Research 3 robot.

| | Human Data | | Robot Data | |
|---|---|---|---|---|
| Task | # Demos / Collect Time (min) / Reset Time (min) | Efficiency (#/s) (Collect/Reset/Total) | # Demos / Collect Time (min) / Reset Time (min) | Efficiency (#/s) (Collect/Reset/Total) |
| PWB | 664 / 33.81 / 12.17 | 3.06 / 1.10 / 4.16 | 192 / 33.92 / 45.44 | 10.60 / 14.20 / 24.80 |
| PTT | 635 / 59.20 / 16.19 | 5.59 / 1.53 / 7.12 | 408 / 95.94 / 106.35 | 14.11 / 15.64 / 29.75 |
| SRP | 209 / 29.34 / 7.38 | 8.42 / 2.12 / 10.54 | 207 / 97.70 / 67.03 | 28.32 / 19.43 / 47.75 |
| SPC | 460 / 44.50 / 19.40 | 5.80 / 2.53 / 8.33 | 196 / 73.04 / 58.80 | 22.36 / 18.00 / 40.36 |
| **Total** | **1968 / 166.85 / 55.14** | **5.09 / 1.68 / 6.77** | **1003 / 300.60 / 277.62** | **17.98 / 16.61 / 34.59** |

**Table 1:** Statistics of the collected demonstration data. We report the number of demonstrations (# Demos), time cost in minutes, and the data collection efficiency in demonstrations per second (#/s). Efficiency is reported for the collection phase, reset phase, and the total process.

**Evaluation Metrics and Protocol.** To ensure a fair and reproducible evaluation, we used standardized metrics and a rigorous testing protocol. While the overall testing environment was kept consistent across all trials, we introduced controlled variations in initial object positions and orientations to robustly assess policy performance. We employ two types of metrics to measure performance.

*Success Rate* (SR): For the short-horizon tasks (*pick up the water bottle* and *pick up the tomato and put it in the tray*), we use the binary success rate. A trial is considered a success only if the robot fully completes the task as described by the language instruction. The final rate is calculated as the total number of successful trials divided by the total number of evaluation trials.

*Task Progress* (TP): For the more complex, long-horizon tasks (*stack the rings on the pillar* and *stack the paper cups*), a simple binary success metric can be too sparse. We therefore measure task progress to provide a finer-grained evaluation. Each task is decomposed into 8 key waypoints, and the policy earns one point for each waypoint it successfully completes. The final score is the average number of completed waypoints across all trials, normalized by the total of 8 waypoints. The detailed waypoint definitions are provided in Appendix A.4.2.

*Evaluation Protocol in Real-World Tasks.* Each task was evaluated over 50 trials with varied initial object configurations (position, orientation) and the introduction of distractors to test for generalization. Specific randomization details for each task are provided in Appendix A.4.1.

### 4.1 PERFORMANCE ANALYSIS

**Baseline.** We establish a strong baseline using the $\pi_0$ architecture (Black et al.), initialized with its publicly available pre-trained weights. In line with standard VLA evaluation (Cadene et al., 2024), we then fine-tune this model for each downstream task using our collected robot demonstrations. To ensure a fair and direct comparison, the amount of robot data used for fine-tuning the baseline is

| Model Variants | SH (SR %) | | LH (TP %) | | Avg. Improvement | |
|---|---|---|---|---|---|---|
| | PWB | PTT | SRP | SPC | SH (SR %) | LH (TP %) |
| Baseline ($\pi_0$) | 48 | 50 | 23.75 | 37.75 | — | — |
| + Trajectory Expert | 58 | 60 | 33.50 | 54.25 | +10.00 | +13.13 |
| + Traj. Expert + Human Data | **76** | **76** | **44.75** | **61.25** | **+27.00** | **+22.25** |

**Table 2:** Performance comparison. Success Rate (%) for Short-Horizon (**SH**) tasks and Task Progress (%) for Long-Horizon (**LH**) tasks are reported. Baseline represents the pre-trained $\pi_0$ with task-specific finetuning.

identical to the robot data subset used for training our full Traj2Action model. This setup provides a rigorous performance benchmark to precisely measure the benefits introduced by our trajectory-guided architecture and the integration of human data.

**Contribution of Trajectory Expert.** We conduct a two-part ablation study to dissect the contributions of trajectory expert. First, we analyze the impact of our architectural modification. To do this, we evaluate a version of our framework that includes the trajectory expert to baseline but is trained solely on robot data. As shown in Table 2, simply adding the trajectory expert improves performance. The benefit is most pronounced on the long-horizon SPC task, where this architectural change alone boosts the Task Progress score from 37.75% to 54.25% (+16.50%). We attribute this to the explicit decomposition of the policy: the trajectory expert acts as a high-level planner by generating a coarse spatio-temporal plan. This plan serves as a robust prior that simplifies the problem for the action expert, allowing it to focus on refining the motion into precise, low-level actions.

Second, we investigate if human data is beneficial without our unified representation. We test this by naively co-training the action expert directly on both human and robot data (i.e., without the trajectory expert) by padding their action label to the same dimension. This approach yields only a marginal improvement over the baseline, achieving a 52% success rate on the *pick up the tomato and put it in the tray* task compared to the baseline's 50%. This minimal 2% gain starkly contrasts with the +10% improvement from our full Traj2Action model. This result critically demonstrates that simply adding human data is insufficient. The unified trajectory space is the essential component that successfully bridges the embodiment gap and unlocks the value of human demonstrations for robot learning.

**Contribution of Human Data.** We then analyze the impact of incorporating human demonstrations for robot learning. As shown in Table 2, integrating human data provides a further, substantial performance boost across all tasks. Compared to the baseline, our full Traj2Action model achieves significant gains: +28% on the *pick up the water bottle* task (reaching 76.00%), +26.00% on the *pick up the tomato and put it in the tray* task (reaching 76.00%), +23.50% on the *stack the paper cups* task (reaching 61.25%), and +21.00% on the *stack the rings on the pillar* task (reaching 44.75%). This demonstrates that long-horizon tasks, in particular, benefit from the diversity of human data. This is because human demonstrators can intuitively execute a wider range of motions—often involving agile re-orientations or angles that are awkward and difficult to perform via teleoperation. By supplementing the robot dataset with these more varied human demonstrations, the trajectory expert learns to predict more robust and generalizable plans. Consequently, the action expert's performance is boosted. This effect is also qualitatively illustrated in Figure 4. In the long-horizon *stack the rings on the pillar* task, the baseline's poor long-term plan causes it to get trapped. In contrast, our policy predicts a coherent trajectory and executes the task successfully. Furthermore, our policy demonstrates superior robustness: after an initial error in the *pick up the tomato and put it in the tray* task, it successfully re-plans and recovers, unlike the baseline which enters a non-productive loop.

Comparative video demos of the baseline and our Traj2Action method are provided in the supplementary material. Furthermore, additional videos featuring detailed analyses are available on our website https://anonymous.4open.science/w/Traj2Action-4A45/.

## 4.2 ABLATION STUDIES

To dissect the contributions of our framework's key components, we conduct a series of ablation studies. We first analyze the impact of scaling the human demonstration dataset and then investigate the properties of the trajectory representation.

**Impact of Human Data Scale.** To quantify how the volume of human data influences policy performance, we vary the number of human demonstrations used in training. As presented in Figure 5,

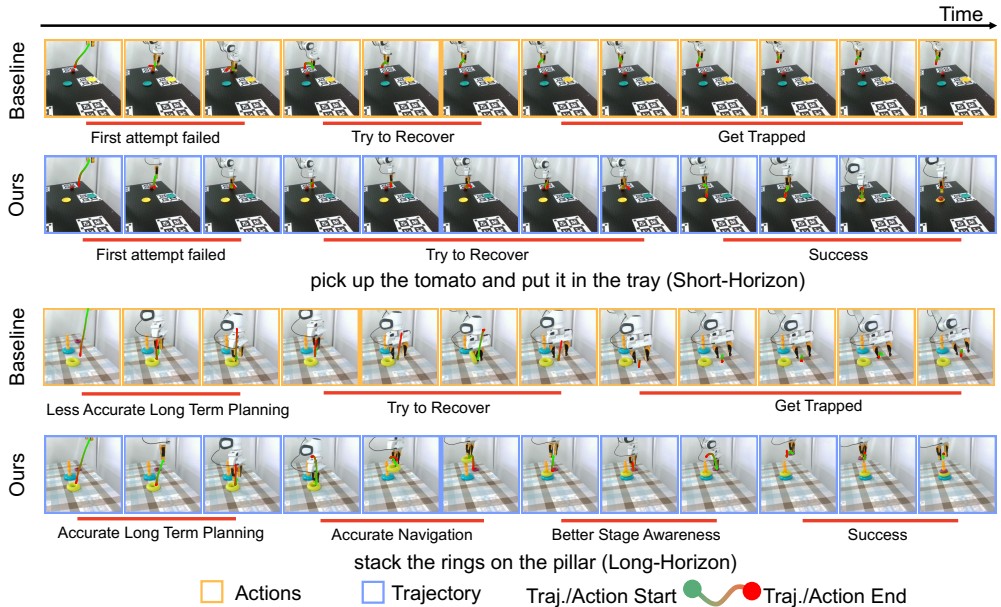

**Figure 4:** Visual comparison of trajectory and action prediction of short- and long-horizon tasks *pick up the tomato and put it in the tray* (top) and *stack the rings on the pillar* (bottom), respectively.

the results show a strong positive correlation between the amount of human data and the final performance across both task types. For the *pick up the tomato and put it in the tray* task (left panel), the Success Rate steadily improves from a 68% baseline (no human data) to a 76% peak with 460 demonstrations. This effect is even more pronounced for the more complex *stack the paper cups* task (right panel), where just 264 human demonstrations catapult the Task Progress score from 37.75% to 57.50%. Using the full 460 demonstrations further boosts performance to 61.25%. This clear scaling effect empirically validates our central hypothesis: human demonstrations are a rich and effective data source for robotic manipulation, and our model adeptly leverages this data to enhance its operational competence.

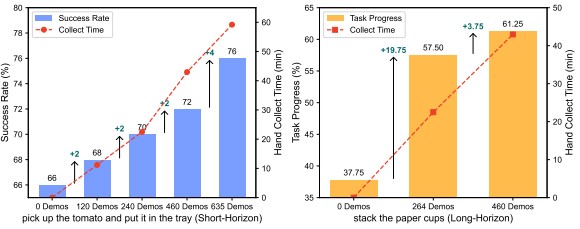

**Figure 5:** Impact of Human Data Scale on Policy Performance. The chart displays the performance on the *pick up the tomato and put it in the tray* task (left) and the *stack the paper cups* task (right) as a function of the number of human demonstrations used in training.

| Strategy | Robot Data (#/min) | Human Data (#/min) | Total Data Time (min) | Performance SR(%) |
|---|---|---|---|---|
| Baseline | 408/202.29 | – | 202.29 | 50 |
| + Trajectory Expert + Robot Data-Only | 408/202.29 | 0/0 | 202.29 | 60 |
| + Trajectory Expert + Human & Robot Data | 270/133.70 | 120/14.25 | 147.95 | 58 |
| | | 240/28.61 | 162.31 | 60 |
| | | 635/75.39 | 209.09 | 62 |

**Table 3:** Performance comparison for the *pick up the tomato and put it in the tray* task under different data collection strategies. The Robot and Human Data columns show the number of demonstrations collected and the total time required in minutes.

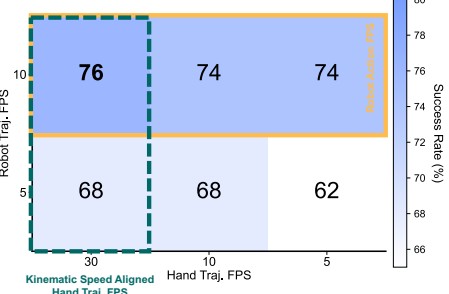

**Figure 6:** Ablation study on the effect of different trajectory sampling frequencies (FPS) on model performance for the *pick up the tomato and put it in the tray* task. The x-axis represents the sampling frequency of the human demonstration trajectory (Hand Traj. FPS), and the y-axis represents the frequency of the robot's predicted trajectory (Robot Traj. FPS). Each cell shows the final Success Rate (%), with darker blue indicating higher performance.

**Human Data act as an Effective Substitute for Robot Data.** To investigate whether labor-saving collected human data can replace time-consuming robot data without degrading policy training performance, we conduct an experiment with three configurations as presented in Table 3.

The results show that human data can be leveraged to train robot policy efficiently. By leveraging just 240 human demonstrations and less robot data, our policy achieves an identical 60% performance compared with model trained solely by robot data, but with a 20% data collection time reduction.

Furthermore, when we increase the human data to 635 demonstrations—bringing the total collection time to a level comparable with the robot-only setup—the performance surpasses the baseline, reaching 62%. Even when the human data is reduced to only 120 demonstrations, the policy still achieves a strong 58% success rate, nearly matching the robot-only performance with a fraction of the data. This clearly shows that a substantial volume of expensive robot data can be effectively replaced by a larger quantity of human data to either decrease data collection cost or boost performance.

This advantage is amplified when considering the overall financial and operational costs, especially for large-scale parallel data collection. While robot data acquisition necessitates expensive, specialized hardware (i.e., the robot arm itself) and skilled teleoperators, human data collection requires only commodity cameras and can be performed by non-expert demonstrators. Therefore, human data is an exceptionally scalable and economically viable solution for learning capable robot policies, significantly lowering the financial and operational barriers to large-scale data collection.

**Impact of Trajectory Sampling Frequency.** A critical challenge in unifying human and robot data is the inherent mismatch in their kinematic speeds; human movements are typically much faster than robot motions. We hypothesized that to sample trajectory data from human and robot with a consistent speed profile (i.e., where the spatial distance between consecutive points is similar), it is necessary to sample the faster human motion at a higher frequency than the slower robot motion.

As presented in Figure 6, our results empirically validate this hypothesis. Peak performance (76% Success Rate) was achieved in a 30-10 configuration, where the human trajectory was sampled at 30 FPS and the robot trajectory at 10 FPS. This 3:1 sampling frequency ratio effectively aligns the motion speeds between human and robot, providing trajectories that are more conducive to effective cross-embodiment policy learning.

Besides, higher success rate is achieved when the predicted robot trajectory is also sampled at 10 FPS compared with sampling frequency at 5 FPS. This demonstrates that a mismatch between the planning frequency of the robot trajectory and the execution frequency of the robot actions degrades performance, as the plan becomes a less temporally misaligned guide for the final control.

**Zero-Shot Generalization to Unseen Task.** We test zero-shot capability of our method on the *pick up the tomato and put it in the tray* task. The policy is trained using 213 robot demonstrations (placing the tomato only in the yellow tray) and 635 human demonstrations (placing it in both the yellow and blue trays). We then evaluated the robot policy on its ability to follow the instruction to "put it in the blue tray". In this challenging zero-shot setting, the robot policy achieves a non-zero success rate of 12% (3 success cases over 25 trials). While modest, this result is significant, as a policy that merely overfits would be expected to have a 0% success rate. This outcome provides crucial evidence that the Trajectory Expert, enriched by diverse human data, can generate a plausible trajectory for the unseen goal and guide the Action Expert in synthesizing a successful action sequence. This demonstrates Traj2Action's capability for generalizing beyond the robot's training data.

## 5 CONCLUSION

We introduced Traj2Action to address the challenge of data-hungry robot policy training by transferring skills from cost-efficient human videos. Our method bridges the morphological gap by unifying demonstrations in a shared representation: the 3D trajectory of the operational endpoint. This trajectory serves as a coarse plan to guide a policy, trained via a co-denoising framework, in generating fine-grained robot actions. Traj2Action significantly outperforms baselines trained solely by robot data quantitatively and qualitatively, particularly on long-horizon tasks requiring foresight. We validated that the trajectory prior enhances planning, that performance scales with the amount of human data, and critically, that low-cost human data can effectively substitute for expensive robot data. This highlights a practical path toward more efficient robot learning.

## 6 REPRODUCIBILITY STATEMENT

To ensure the reproducibility of our research, we provide comprehensive details on our methodology, code, and experimental setup.

**Code** All code used for data processing, model training, and evaluation is publicly available in our anonymized repository at https://anonymous.4open.science/r/Traj2Action-4A45/. The repository includes instructions for setting up the environment and running the experiments.

**Data Collection** A detailed description of the hardware and software setup for both the human hand motion capture and the robot data collection systems is provided in Appendix A.2.

**Training** We provide exhaustive details on the training procedures, including hyperparameters, model architectures, and training schedules, in Section 3 and Appendix A.5.

**Evaluation** The standardized protocol used for evaluating our policy across all four manipulation tasks, including the definition of metrics and the setup for initial state randomization, is detailed in Section 4 and Appendix A.4.

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

# A APPENDIX

## A.1 STATEMENT ON THE USE OF LARGE LANGUAGE MODELS (LLMS)

In accordance with the ICLR 2026 policy, we disclose the use of a Large Language Model (LLM) in the preparation of this manuscript. The LLM was employed exclusively as a tool for language polishing, including grammar correction, style improvement, and enhancement of readability. The core scientific contributions, including all research ideas, methodologies, experimental designs, data analysis, and conclusions, were conceived and executed entirely by the human authors. The LLM did not contribute to any substantive aspect of the research. The authors assume full responsibility for all content presented in this paper.

## A.2 DATA COLLECTION SYSTEM DETAILS

This appendix provides a comprehensive description of the hardware and software components used in our human and robot data collection systems.

### A.2.1 HUMAN HAND MOTION CAPTURE SYSTEM

Our vision-based motion capture system is designed to reconstruct detailed 3D hand motions from multi-view images.

**Hardware Setup.** The system is built around four synchronized cameras operating at 30 Hz (as shown in Figure 2). Three of these are static industrial cameras mounted on a rigid frame surrounding the data collection workspace, providing top-down, side-left, and side-right perspectives of the user's hand. This arrangement is critical for minimizing self-occlusion. The fourth is a lightweight camera mounted on the back of the user's hand, providing an ego-centric view analogous to the wrist camera on our robot.

**3D Hand Pose Estimation Pipeline.** The reconstruction pipeline is implemented as a two-stage process for each timestamped set of images.

- 2D Keypoint Detection: We leverage the robust Google MediaPipe (Zhang et al., 2020) Hand Landmarker to process each video stream independently. For each hand detected in an image, the model outputs 21 2D landmarks $(\mathbf{u}, \mathbf{v})$ in pixel coordinates. These landmarks serve as the 2D evidence for our 3D reconstruction.

- Model-Based 3D Reconstruction: We fit the MANO (Romero et al., 2017) model to these multi-view 2D detections. MANO is a parametric model defined by shape parameters $\boldsymbol{\beta} \in \mathbb{R}^{10}$, pose parameters $\boldsymbol{\theta} \in \mathbb{R}^{45}$ (representing the axis-angle rotations of 15 joints), a global orientation $\mathbf{R} \in SO(3)$, and a global translation $\mathbf{t} \in \mathbb{R}^3$.

The fitting is formulated as an optimization problem where we seek the parameters that minimize the reprojection error. The objective function is the mean squared L2 error between the projected 3D MANO joints and the 2D keypoints detected by MediaPipe:

$$(\boldsymbol{\theta}^*, \mathbf{R}^*, \mathbf{t}^*) = \arg\min_{\boldsymbol{\theta}, \mathbf{R}, \mathbf{t}} \frac{1}{N_{\text{cams}}} \sum_{c=1}^{N_{\text{cams}}} \frac{1}{21} \sum_{j=1}^{21} \|\Pi(\mathbf{K}_c, [\mathbf{R}_c | \mathbf{t}_c], \mathbf{J}_j(\boldsymbol{\beta}, \boldsymbol{\theta}, \mathbf{R}, \mathbf{t})) - \mathbf{k}_{c,j}\|_2^2$$

Here, $\mathbf{J}_j(\cdot)$ is the 3D location of the j-th joint generated by the MANO model, $\Pi(\cdot)$ is the camera projection function, and $\mathbf{k}_{c,j}$ is the corresponding detected 2D keypoint in camera c.

As detailed in our provided source code, this optimization problem is solved iteratively using the Adam optimizer. For any given subject, the shape parameter $\boldsymbol{\beta}$ is fixed to the mean shape ($\boldsymbol{\beta} = \mathbf{0}$) to ensure temporal consistency. We initialize the hand in a neutral pose and optimize the pose ($\boldsymbol{\theta}$), global orientation ($\mathbf{R}$), and translation ($\mathbf{t}$) parameters for each frame. The learning rate is managed by a Cosine Annealing scheduler, starting at 0.14 and decaying to 0.08. To balance accuracy and computational load, we employ a dynamic iteration scheduler that reduces the number of optimization steps for later frames in a sequence, assuming smaller inter-frame motion. This process yields a complete, kinematically consistent 3D representation of the hand pose. The full algorithm is summarized in Algorithm 1.

---

**Algorithm 1** Model-Based 3D Hand Reconstruction

---

**Require:**
    Observed 2D keypoints $\{\mathbf{k}_{c,j}\}$, camera parameters $\{\mathbf{K}_c, [\mathbf{R}_c|\mathbf{t}_c]\}$, and the MANO model $\mathcal{M}$.
**Ensure:**
    Optimized parameters $\boldsymbol{\theta}^*, \mathbf{R}^*, \mathbf{t}^*$.

1: **procedure** RECONSTRUCTHANDPOSE($\{\mathbf{k}_{c,j}\}, \dots$)
2:     $\boldsymbol{\theta} \leftarrow \mathbf{0}, \mathbf{R} \leftarrow \mathbf{I}, \mathbf{t} \leftarrow \mathbf{0}$                ▷▷ Initialize optimizable parameters
3:     $\boldsymbol{\beta} \leftarrow \mathbf{0}$                                   ▷▷ Use fixed mean hand shape
4:     $N_{iter} \leftarrow$ ITERATIONSCHEDULER.GET_ITERATIONS
5:     **for** $i = 1 \rightarrow N_{iter}$ **do**
6:         $\{\mathbf{J}_j\} \leftarrow \mathcal{M}(\boldsymbol{\beta}, \boldsymbol{\theta}, \mathbf{R}, \mathbf{t})$         ▷▷ Generate 3D joints via MANO forward pass
7:         $loss \leftarrow \frac{1}{N_{\text{cams}} \cdot 21} \sum_{c,j} \|\Pi(\mathbf{K}_c, [\mathbf{R}_c|\mathbf{t}_c], \mathbf{J}_j) - \mathbf{k}_{c,j}\|_2^2$     ▷▷ Compute mean squared reprojection error
8:         Update parameters $\boldsymbol{\theta}, \mathbf{R}, \mathbf{t}$ via gradient descent on $loss$.
9:     **end for**
10:     **return** $\boldsymbol{\theta}, \mathbf{R}, \mathbf{t}$ as $\boldsymbol{\theta}^*, \mathbf{R}^*, \mathbf{t}^*$
11: **end procedure**

---

**Computational Hardware.** The 2D keypoint detection and, more significantly, the iterative 3D model optimization are computationally intensive. All above-mentioned operations, including both the MediaPipe inference and the MANO model fitting, are accelerated on an NVIDIA GeForce RTX 4090 GPU to ensure efficient data processing.

### A.2.2 ROBOT DATA COLLECTION SYSTEM

**Hardware Setup.** The robotic setup consists of a Franka Research 3 arm with a UMI Gripper (Chi et al., 2024) as the end-effector. For intuitive control, we employ a 6-DoF SpaceMouse as the primary input device for the operator. The visual data acquisition system includes two synchronized RGB-D cameras. Including a static main camera positioned to the robot's left (called Left Camera) and a wrist-mounted camera providing an ego-centric view (called Ego Camera). Notably, since it is easy to be occluded by the robot arm, the Top Camera is not used in the robot data collection system.

**Control and Data Recording.** The operator directly controls the robot by manipulating the SpaceMouse. The inputs from the device are mapped to linear and angular velocity commands $(\dot{x}, \dot{y}, \dot{z}, \dot{\omega}_x, \dot{\omega}_y, \dot{\omega}_z)$ that dictate the motion of the robot's end-effector in Cartesian space. This velocity-based control scheme allows for smooth and precise execution of tasks. During each demonstration, we record multiple synchronized data streams at a frequency of 30 Hz, including robot proprioceptive data, operator control commands, and multi-view visual data.

**Initial State Randomization.** To ensure the learned policy is robust to perturbations, we introduce randomization to the robot's starting pose for each demonstration. Instead of starting from a single, fixed position, the end-effector is initialized at a random offset from a canonical start pose. This offset is sampled from a uniform distribution within a predefined Cartesian volume. By exposing the policy to a diverse set of initial conditions during training, this strategy significantly improves its ability to generalize and recover from states that deviate from the expert trajectories, a crucial capability for real-world deployment.

### A.3 CAMERA CALIBRATION

To accurately and consistently determine the extrinsic parameters (i.e., the relative 3D poses) of our multi-camera system, we designed a custom calibration target integrated directly into the experimental platform. The system consists of five fiducial markers placed at fixed locations on the workspace surface. We use AprilTag markers from the `t36h11` dictionary, with a tag size of 70 mm and an inter-tag spacing of 21 mm. The spatial distribution of these five tags is strategically designed to guarantee that any camera in our setup can simultaneously observe at least two markers, a critical requirement for robust extrinsic calibration with the Kalibr toolbox. This setup allows for quick and

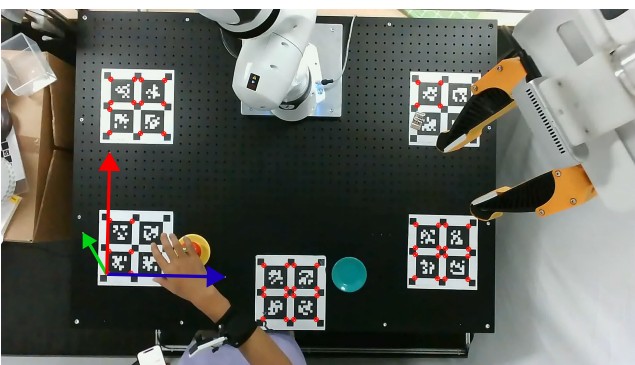

**Figure 7:** Our camera extrinsic calibration setup. Five AprilTag markers are strategically distributed across the robot's workspace, ensuring that multiple tags are visible from any camera viewpoint for robust pose estimation.

reliable calibration, yielding a positioning precision that fully meets the demands of our manipulation experiments.

### A.4 EVALUATION METRICS AND PROTOCOL

#### A.4.1 EVALUATION PROTOCOL DETAILS

For each task, a full evaluation round consists of 50 trials with varied initial object configurations to test for generalization.

- For *pick up the water bottle*, we pre-selected 50 distinct bottle positions and orientations uniformly across the workspace. These poses were marked with invisible ink to ensure consistent placement during evaluation.

- For *pick up the tomato and put it in the tray*, we defined 24 distinct triplets of positions for the tomato, yellow tray, and blue tray. For each triplet, we tested two separate prompts ("put it in the yellow tray" and "put it in the blue tray"), resulting in 48 trials. For the final triplet configuration, we added a pumpkin as a distractor object and tested the two prompts again, yielding 2 more trials for a total of 50.

- For *stack the rings on the pillar*, we selected 50 different position triplets for the pillar, the yellow ring, and the red ring to serve as the initial scene for each of the 50 trials.

- For *stack the paper cups*, we similarly selected 50 different position triplets for the three individual paper cups.

#### A.4.2 WAYPOINT DEFINITIONS FOR TASK PROGRESS

The 8-step waypoint decomposition for the task progress metric is defined in Table 4.

#### A.4.3 EVALUATION SYSTEM

**Hardware Setup.** The evaluation system mirrors the data collection setup, utilizing the same Franka Research 3 robot arm with a UMI Gripper end-effector. The visual input is provided by two synchronized RGB-D cameras: a static Left Camera positioned to the robot's left and a wrist-mounted Ego Camera. This configuration ensures that the policy receives consistent visual information during both training and evaluation.

**Policy Inference Strategy.** In each cycle, the system first captures a comprehensive observation from its cameras and the robot's current physical state. This observation is then fed into the learned policy ($\pi_\theta$), which makes a decision by predicting an entire sequence of future actions. Finally, the system enters the action execution phase, stepping through the predicted sequence and sending individual motion and gripper commands to the robot at a fixed frequency until the sequence is complete, at which point the loop repeats. The complete algorithm is summarized in Algorithm 2.

| stack the rings on the pillar | stack the paper cups |
|---|---|
| (1) The gripper makes contact with the yellow ring. | (1) The gripper makes contact with the first cup. |
| (2) The gripper successfully grasps the yellow ring. | (2) The gripper successfully grasps the first cup. |
| (3) The gripper, holding the ring, makes contact with the pillar. | (3) The gripper, holding the first cup, makes contact with the second cup. |
| (4) The gripper successfully places the yellow ring onto the pillar. | (4) The gripper successfully places the first cup into the second cup. |
| (5) The gripper makes contact with the red ring. | (5) The gripper makes contact with the stacked (second) cup. |
| (6) The gripper successfully grasps the red ring. | (6) The gripper successfully grasps the stack of two cups. |
| (7) The gripper, holding the ring, makes contact with the pillar. | (7) The gripper, holding the stack, makes contact with the third cup. |
| (8) The gripper successfully places the red ring onto the pillar. | (8) The gripper successfully places the stack of two cups into the third cup. |

Table 4: Waypoint definitions for the two Long-Horizon tasks.

---

**Algorithm 2** Core Robot Control Loop

---

**Require:** A learned policy $\pi_\theta$, a robot interface $\mathcal{R}$, and camera interfaces $\mathcal{C}$.
1:  **procedure** EXECUTEPOLICY($\pi_\theta, \mathcal{R}, \mathcal{C}$)
2:     STARTCAMERASTREAMS($\mathcal{C}$)
3:     $S_{\text{gripper}} \leftarrow$ {current: OPEN, target: OPEN}
4:     **while** not STOPSIGNALRECEIVED **do**
5:        $O \leftarrow$ GETOBSERVATION($\mathcal{R}, \mathcal{C}$)                                    ▷ Perception
6:        $A_{\text{chunk}} \leftarrow \pi_\theta(O)$                                              ▷ Decision
7:        **for all** action $a$ in $A_{\text{chunk}}$ **do**                          ▷ Action Execution
8:           $a_{\text{motion}}, a_{\text{gripper}} \leftarrow$ DecomposeAction($a$)
9:           EXECUTEMOTIONASYNC($\mathcal{R}, a_{\text{motion}}$)
10:          **if** $a_{\text{gripper}} > 0.95$ **then** $S_{\text{gripper}}$.target $\leftarrow$ CLOSED
11:          **else if** $a_{\text{gripper}} < -0.95$ **then** $S_{\text{gripper}}$.target $\leftarrow$ OPEN
12:          **end if**
13:          **if** $S_{\text{gripper}}$.current $\neq S_{\text{gripper}}$.target **then**
14:             ACTUATEGRIPPERASYNC($\mathcal{R}, S_{\text{gripper}}$.target)
15:             $S_{\text{gripper}}$.current $\leftarrow S_{\text{gripper}}$.target
16:          **end if**
17:          SLEEP($\Delta t$)
18:       **end for**
19:    **end while**
20: **end procedure**

---

## A.5   MODEL TRAINING

The model training in this study was carried out using the Hugging Face Accelerate framework for distributed training on 4 NVIDIA H800 GPUs.

We initialize our model using the pre-trained weights from $\pi_0$ (Black et al.), a state-of-the-art vision-language-action model. The vision encoder is kept frozen during training to leverage its pre-learned visual representations. The training process employs the AdamW optimizer with a learning rate of $1 \times 10^{-4}$, and a linear learning rate scheduler with warmup and decay phases. The parameters of trajectory expert ($g_\mathcal{T}$) is set to be the same as action expert ($\pi_0$), except for the output dimension.

The training framework is adapted from LeRobot (Cadene et al., 2024), with modifications to accommodate our multi-view input and dual expert architecture. We utilize a batch size of 32, with each training iteration processing chunks of 50 timesteps for both actions and trajectories.

The detailed hyperparameter configuration is presented in Table 5.

| Category | Parameter & Value |
| --- | --- |
| **Hardware & Framework** | |
| Compute Device | 4x NVIDIA GPU |
| Distributed Training Framework | Hugging Face Accelerate |
| **Data Input & Preprocessing** | |
| Input Features | |
|    Left Camera Image | Shape: [3,224,224] |
|    Ego-view Image | Shape: [3,224,224] |
|    State Vector | Dimension: 8 |
|    State Trajectory | Dimension: 3 |
| Preprocessing | |
|    Image Resizing | [224,224] (with padding) |
|    State/Action/Trajectory Normalization | Mean-Std Normalization |
|    Image Normalization | Identity (no operation) |
| **Model Architecture** | |
| Base Model Type | $\pi_0$ (Black et al.) |
| Freeze Vision Encoder | True |
| Projection Width | 1024 |
| Max State Dimension | 32 |
| Max Action Dimension | 32 |
| Max Trajectory Dimension | 32 |
| **Optimizer & Scheduler** | |
| Optimizer | AdamW |
|    Learning Rate | $1 \times 10^{-4}$ |
|    Betas | (0.9, 0.95) |
|    Epsilon | $1 \times 10^{-8}$ |
|    Weight Decay | $1 \times 10^{-10}$ |
| Learning Rate Scheduler | Linear Warmup and Decay |
|    Warmup Steps | 1000 |
|    Decay Steps | 160,000 |
|    Min Learning Rate | $2.5 \times 10^{-6}$ |
| **Training Details** | |
| Gradient Accumulation Steps | 1 |
| Action Chunk Length | 50 |
| Trajectory Chunk Length | 50 |
| Traj to Action Random Mask Probability | 0.2 |
| **Output Features** | |
|    Actions | Dimension: 7 |
|    Trajectory | Dimension: 3 |

**Table 5:** Model Training Hyperparameter Details

### A.6 ADDITIONAL VISUALIZATION AND QUALITATIVE ANALYSIS.

For the visualization of task *pick up the tomato and put it in the tray* and *stack the rings on the pillar*, please refer to Figure 4. For the visualization of task *pick up the water bottle* and *stack the paper cups*, please refer to Figure 8.

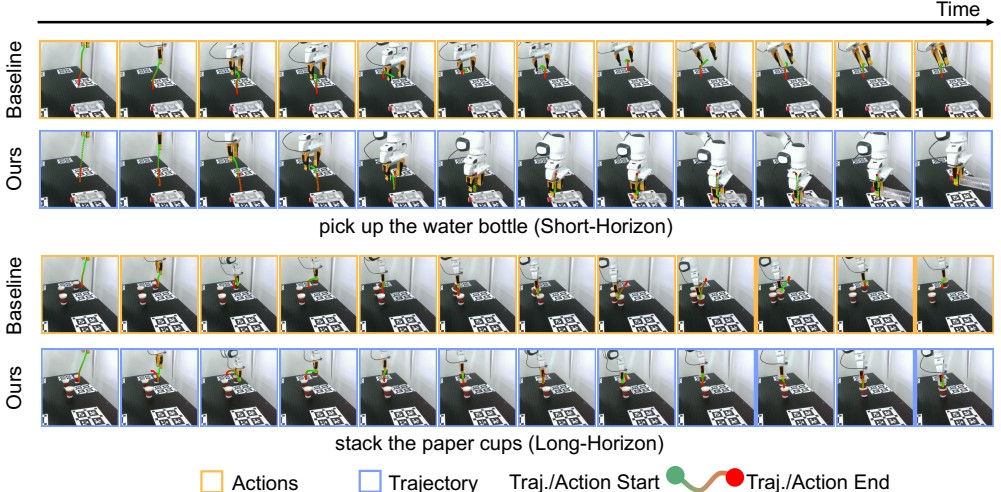

**Figure 8:** Visual comparison of trajectory and action predicition of selected tasks *pick up the water bottle* (top) and *stack the paper cups* (bottom).