# OpenReview forum: "From Human Hands to Robot Arms: Manipulation Skills Transfer via Trajectory Alignment"
_ICLR.cc/2026/Conference — ICLR 2026 Conference Withdrawn Submission_

### Official Review · Reviewer_89t7 · 2025-10-20

**Soundness:** 3
**Presentation:** 3
**Contribution:** 3
**Rating:** 6
**Confidence:** 4

**Summary:**

This paper proposes Traj2Action, a simple two-expert policy for transferring manipulation skills from human demonstrations to robots. It uses the 3D end-effector/hand trajectory (positions only) as a unified intermediate representation, first predicting a coarse trajectory from multi-view images and language, then conditioning a robot-only action expert to output precise deltas (position, rotation, gripper) via a joint co-denoising / flow-matching objective. On four Franka tasks, the method reports sizable gains over a VLA baseline and indicates human data can partially substitute for teleoperated robot data, with ablations on human-data scale and trajectory sampling frequency.

**Strengths:**

- Conceptual clarity & simplicity. Using a position-only trajectory as the embodiment-agnostic “common divisor” is clean and broadly applicable; the coarse-to-fine decomposition is intuitive.
- Empirical signal. Real-robot evaluations show consistent improvements over π₀, especially on long-horizon tasks; scaling human data helps, and ablations on FPS alignment are thoughtful.
- Cost/efficiency angle. Evidence that human demos can replace a portion of costly robot data is valuable for the community’s data economics.

**Weaknesses:**

- Limited baselines beyond ablations. Most results compare to π₀ with or without the trajectory expert; there is no direct comparison to recent human-robot co-training / cross-embodiment approaches (e.g., egocentric human-video VLA pretraining, video-track-to-action, motion-track alignment, etc.). This makes it hard to judge progress relative to state-of-the-art alternatives designed for the same setting.
- Representation scope. The unified trajectory uses positions only; orientation/grasp semantics are pushed entirely to the action expert. It remains unclear when this abstraction is sufficient (e.g., tool use, in-hand reorientation) and when it fails. A task-type breakdown would clarify boundary conditions.
- Reporting and statistics. Many deltas are reported without confidence intervals/significance tests across seeds or days; robustness to camera calibration errors and to workspace/domain shifts is not deeply probed.

**Questions:**

- Comparative baselines. Can you add direct comparisons to human-robot co-training or egocentric video VLA approaches and trajectory/track-based transfer (e.g., track-to-act / motion-tracks-style methods)? Matching robot-data budgets would strengthen claims.
- When do positions suffice? For tasks requiring tight orientation control (e.g., peg-in-hole, tool insertion), does a position-only unified space degrade?

---

### Official Review · Reviewer_NXTf · 2025-10-26

**Soundness:** 2
**Presentation:** 3
**Contribution:** 1
**Rating:** 0
**Confidence:** 5

**Summary:**

This papers proposes Traj2Action, a robot manipulation method to use a pose trajectory as a shared action space for both grippers and human hand. The authors demonstrate that this framework, when using human data, can have a large performance gain over short/long horizon tasks.

**Strengths:**

1. The paper overall is very easy to follow
2. The human data experiment is very motivating, it both demonstrates that performance can be scaled up even with human data, and human data can lead to more robust policy.
3. The real-world experiment results are convincing

**Weaknesses:**

- My main concern is novelty. The design and benefits of intermediate action spaces have been thoroughly studied in numerous papers (not novel). Even when we consider human to robot transfer, MimicPlay (CoRL 2023) already have very similar problem setup and framework, except that this paper is based on a VLM backbone, which is not available at the time. There are also already many papers focusing on human to dexterous hand transfer, where a shared action space is naturally the hand pose. Overall I very much appreciate the efforts that went into this framework and system. But I believe the method itself is not significant enough for a venue like ICLR.
- This work focus on human data and robot data collected from a single custom workspace setup. The generalization to larger-scale human videos is questionable.
- The robot tasks are too simple, just pick-and-drop type of tasks.
- The authors only conduct self-comparison in the experiments, which would be strengthened by more comparisons.
- The authors collect over 1000 demonstrations for each task, which seems to be an order of magnitude more than conventional standard (~100), I wonder what is the rationale behind his design choice.

**Questions:**

(in weakness)

---

### Official Review · Reviewer_oZ1F · 2025-10-30

**Soundness:** 2
**Presentation:** 4
**Contribution:** 2
**Rating:** 4
**Confidence:** 5

**Summary:**

This paper addresses the challenge of transferring manipulation skills from human videos to robots by proposing Traj2Action, a framework that bridges the morphological gap between human hands and robot end-effectors using 3D trajectories as a unified representation. The method operates in a coarse-to-fine manner, where a trajectory expert first generates a high-level motion plan from human and robot data, which then guides an action expert to synthesize precise, robot-specific actions via a joint denoising training scheme. Real-world experiments demonstrate that this approach allows higher data collection efficiency and outperforms the $\pi_0$ baseline.

**Strengths:**

1. The paper is well organized and easy to follow.
2. The idea is well motivated.
3. The experiment results show that including human data improves the performance.

**Weaknesses:**

1. The method relies on a multi-camera motion capture system with a wrist-mounted ego-view camera to collect human data. This setup is far from scalable. It is impossible to collect in-the-wild data or even learn from internet videos.
2. The representation of the human hand action as translations of a single 3D point (the thumb-index midpoint) is a significant oversimplification. This abstraction, while effective for the pick-and-place tasks presented, inherently limits the method's applicability. It cannot capture orientation cues from the human wrist or the complex finger coordination required for rotation-critical tasks (e.g., screwing) or contact-rich, dexterous manipulations (e.g., in-hand manipulation, tool using). Consequently, the experimental validation on four variants of basic pick-and-place tasks does not demonstrate the method's efficacy for the broader spectrum of robotic manipulation.
3. The chosen tasks (PWB, PTT, SRP, SPC) are all structural variations of pick-and-place conducted in a controlled table-top setting.
4. The paper only compares against $\pi_0$, but lacks direct comparisons to other recent SOTA cross-embodiment imitation learning methods (e.g., Track2Act, ATM, Im2Flow2Act, EgoMimic).
5. The paper's claims of generalization are weakly supported. The sole generalization experiment is a simple instructional combination generalization ("put it in the blue tray" instead of "yellow"), which achieves a low success rate (12%). Crucially, the work lacks evaluation on other critical axes of generalization essential for real-world deployment, such as scene, object, and motion generalization.

**Questions:**

For my main concerns, please refer to the “Weaknesses” part. Here are some minor questions:

1. The paper states that the Trajectory Expert's weights are initialized from the pretrained Action Expert of the $\pi_0$ model. However, the dimensions differ: actions are 7-dimensional, while trajectories are 3-dimensional. Could the authors clarify the specific architectural modification to accommodate this? Was a simple linear/MLP projection layer added, and if so, how was it initialized?
2. For the human data collection, the ego-centric view is provided by a camera mounted on the back of the user's hand. Given that this cannot be secured as rigidly as on a robot arm, what measures were taken to ensure the camera's pose remained stable relative to the hand? How do you deal with the potential camera shifting or loosening during demonstrations, which could introduce noise into the visual input?

---

### Official Review · Reviewer_yabZ · 2025-11-02

**Soundness:** 2
**Presentation:** 2
**Contribution:** 1
**Rating:** 2
**Confidence:** 5

**Summary:**

The paper proposes Traj2Action, a human→robot skill transfer framework that uses the 3D end-effector trajectory as a unified intermediate representation. A Trajectory Expert (trained on human+robot data) predicts a coarse future trajectory, which then conditions an Action Expert (trained on robot data) to output precise deltas in position, orientation, and gripper state via a joint (flow-matching) co-denoising objective. The authors collect human hand trajectories with multi-view capture and an additional wrist/ego view to reduce observation mismatch. On four real-robot tasks (two short-horizon, two long-horizon), Traj2Action improves over a pi_0 baseline, shows scaling gains with more human data, and demonstrates that human demos can replace a sizable fraction of robot demos at comparable performance/cost.

**Strengths:**

The strengths of this work are:

- The problem setting is very presented and motivated: human-to-robot skill transfer is important and practical; focusing on mapping human demos into robot-usable signals is worthwhile.

- They conduct real-robot evaluation. Includes on-hardware tests rather than sim-only results, with clear task metrics (e.g., success rate / waypoint progress).

**Weaknesses:**

The main weaknesses are limited methodological novelty and unfair experimental comparisons.

-  Limited novelty and missing related work in the part of translating human motion to robot: The proposed “unified trajectory space” offers little methodological novelty. The paper omits key position-based retargeting literature—e.g., AnyTeleop [1] and works that map human video motion to robot trajectories for downstream learning [2–5]. The claim that fingertip positions as a unified motion representation is not new at all (see [1-5] and lots of other recent works); moreover, the method ignores hand/gripper geometry by directly mapping the human fingertip midpoint to the robot end-effector pose. What happens under a different gripper or finger lengths? This appears to require ad-hoc offsets and re-tuning. Prior work typically optimizes retargeting to minimize pose error under the robot’s kinematics; while such methods are imperfect (they ignore object dynamics), the proposed heuristic is arguably weaker because it ignores the robot kinematics entirely.

[1]. Qin, Yuzhe, et al. "Anyteleop: A general vision-based dexterous robot arm-hand teleoperation system." arXiv preprint arXiv:2307.04577 (2023).

[2]. Shaw, Kenneth, Shikhar Bahl, and Deepak Pathak. "Videodex: Learning dexterity from internet videos." Conference on Robot Learning. PMLR, 2023.

[3]. Liu, Yangcen, et al. "Immimic: Cross-domain imitation from human videos via mapping and interpolation." arXiv preprint arXiv:2509.10952 (2025).

[4]. Chen, Zoey Qiuyu, et al. "Dextransfer: Real world multi-fingered dexterous grasping with minimal human demonstrations." arXiv preprint arXiv:2209.14284 (2022).

[5]. Bahl, Shikhar, Abhinav Gupta, and Deepak Pathak. "Human-to-robot imitation in the wild." arXiv preprint arXiv:2207.09450 (2022).

- Limited novelty and missing related work in the part of the policy learning: Training the “trajectory expert” on both human and robot data resembles standard co-training strategy [6]. It’s also unclear whether you use an explicit co-training scheme or simply merge datasets. The “action expert,” conditioned on the trajectory expert and trained on robot-only data, amounts to robot-data fine-tuning for higher precision—similar in spirit to prior pipelines (e.g., [2]). Beyond fitting this into a VLA-style backbone, it’s not clear what is conceptually new.

[6]. Maddukuri, Abhiram, et al. "Sim-and-real co-training: A simple recipe for vision-based robotic manipulation." arXiv preprint arXiv:2503.24361 (2025).

- Evaluation design and baseline fairness: VLA methods target open-world, diverse task sets, but your evaluation focuses on a small set of pick-and-place-like tasks derived from human demos. Yet you primarily compare against VLA baselines rather than recent human-demo learning methods. Therefore, unless your tasks align with the breadth and diversity of standard VLA benchmarks, comparing only to VLA is not a fair test of your contributions.

**Questions:**

See Weaknesses.

---

### Note · Authors · 2026-01-20

I have read and agree with the venue's withdrawal policy on behalf of myself and my co-authors.